# Diabetic Neuropathy of the Retina and Inflammation: Perspectives

**DOI:** 10.3390/ijms24119166

**Published:** 2023-05-23

**Authors:** Guzel Bikbova, Toshiyuki Oshitari, Mukharram Bikbov

**Affiliations:** 1Department of Ophthalmology and Visual Science, Graduate School of Medicine, Chiba University, Inohana 1-8-1, Chuo-ku, Chiba 260-8670, Japan; gbikbova@gmail.com; 2Ufa Eye Research Institute, Pushkin Street 90, Ufa 450077, Russia; 3Department of Ophthalmology, School of Medicine, International University of Health and Welfare, 4-3 Kozunomori, Narita 286-8686, Japan

**Keywords:** diabetes, hyperglycemia, dyslipidemia, inflammation, atherosclerosis

## Abstract

A clear connection exists between diabetes and atherosclerotic cardiovascular disease. Consequently, therapeutic approaches that target both diseases are needed. Clinical trials are currently underway to explore the roles of obesity, adipose tissue, gut microbiota, and pancreatic beta cell function in diabetes. Inflammation plays a key role in diabetes pathophysiology and associated metabolic disorders; thus, interest has increased in targeting inflammation to prevent and control diabetes. Diabetic retinopathy is known as a neurodegenerative and vascular disease that occurs after some years of poorly controlled diabetes. However, increasing evidence points to inflammation as a key figure in diabetes-associated retinal complications. Interconnected molecular pathways, such as oxidative stress, and the formation of advanced glycation end-products, are known to contribute to the inflammatory response. This review describes the possible mechanisms of the metabolic changes in diabetes that involve inflammatory pathways.

## 1. Introduction

Diabetes is a metabolic disorder affecting the human body’s ability to regulate glucose. Chronic hyperglycemia, dyslipidemia, insulin resistance, and glyco/lipoxidation end-products are the mediators of diabetes and the main clinical and diagnostic features of diabetes that cause vascular complications. Elevated glucose associated with diabetes can result in dysfunction and failure of many organs, such as the retina (resulting in visual acuity loss and blindness), kidneys, nerves, and heart [1]. Diabetic neuropathies characterized by a progressive loss of nerve fibers are common complications affecting approximately 50% of patients with diabetes [2]. The estimated prevalence of diabetes will rise to 540 million people in 2021 and 780 million people in 2045 [3]. Among them, 95 million diabetic patients have already developed diabetic retinopathy in the world [4]. In Japan, 23.5% of diabetic patients had diabetic retinopathy in 2014 [5]. Most younger patients tend to have poor glycemic control because of their Western lifestyle, which will increase the prevalence of diabetic retinopathy in younger patients in the future in Japan [5]. Diabetic retinopathy is defined as a tissue-specific neurovascular impairment of multiple types of cells, including neurons, glial cells, and vascular cells in the retina. Among these cells, abnormalities of neuronal cells, i.e., neuronal cell death and axonal degeneration, are directly related to vision loss in diabetic patients. Thus, understanding the pathophysiology of neurovascular impairment in diabetic retinopathy is needed for the complete management of diabetic retinopathy in clinical practice.

Various hypotheses have been suggested to explain the mechanisms by which complications develop in diabetes. Well-recognized risk factors such as a sedentary lifestyle, lack of physical activity, and energy-dense diets are considered major but are modifiable through behavioral and environmental changes [6,7]. Genetic predisposition, ethnicity, and aging are unmodifiable risk factors for type 2 diabetes. Increasing evidence has shown that inflammatory pathways are the principal, common pathogenetic mediators in the natural course of diabetes under the stimulus of these risk factors [8]. Herein, we highlight the role of inflammation in the pathophysiology of diabetes and diabetic retinopathy. This review provides an overview of the current knowledge on pathogenetic changes involved in inflammation in diabetes and diabetic retinopathy, along with perspectives on future anti-inflammatory therapies for diabetes and diabetic retinopathy.

## 2. Historical Perspectives

Studies show that the authority of science is currently being questioned. Such historical errors as the vilification of cholesterol and the promotion of sugar have caused irreparable damage to the health of generations. Kearns et al. analyzed newly released historical documents suggesting that nearly five decades of research on nutrition and heart disease, including many of today’s dietary recommendations, may have been largely influenced by the sugar industry, which paid scientists in the 1960s to downplay the link between sugar and heart disease and instead promote saturated fat as the culprit in heart disease [9]. Dyslipidemias are major causes of cardiovascular morbidity and mortality worldwide [10]. Historically, since pathologists first introduced the term arteriosclerosis in the 19th century, the passive accumulation of cholesterol into the arterial walls has been believed to cause atherosclerosis. Atherosclerosis was primarily thought to be a lipid disorder for much of the 20th century [11].

### 2.1. Dyslipidemia and Hyperglycemia

The current literature does not support the idea that dietary cholesterol increases the risk of heart disease in healthy individuals, and recent studies showed no association between dietary cholesterol (egg consumption) and serum cholesterol [12]. Hyperglycemia is often associated with increased intestinal lipoproteins and reduced high-density lipoprotein (HDL) resulting from hyperglycemia-induced glycation of lipoproteins by the low-density lipoprotein (LDL) receptor, which reduces lipoprotein uptake and catabolism [13]. A high dietary carbohydrate load increases the glycation of intestinal lipoproteins, prolongs their circulation, and increases their plasma concentration by inhibiting lipoprotein lipase. Circulating advanced glycation end products (AGEs) also bind lipoproteins and delay their clearance, a mechanism implicated in dyslipidemia in diabetic nephropathy [14]. Recent studies have shown relationships between AGE accumulation in tissues and the development of diabetic complications, thus confirming the significant role of AGEs in the development of vascular complications with age-related diseases [15]. Even low AGE concentrations (e.g., 10 μg/mL) were shown to increase neuronal apoptosis in retinal neurons and reduce the burden of regenerated neuritis [15]. Glycation increases the lipoprotein proportions taken up by inflammatory cells and decreases the proportions taken up by hepatocytes; this promotes atheromatous plaque formation and stimulates inflammation. Hyperglycemia increases the formation of oxidized and glycated LDL [13]. A recent study on the role of lipid oxidation in atherosclerosis development provides a graph demonstrating the development of a glucose response after eating, along with increased oxidative stress and altered endothelial function [16].

Ito et al. described the relationship between glucose uptake in the blood and the subsequent increase in circulating inflammatory lipoxidation products [14]. Hyperglycemia impacts patients’ lipid profiles, thus disrupting the LDL/HDL balance, generating AGEs, and activating the AGE/receptor for the AGE (RAGE) pathway [17,18].

Oxidized LDL products provoke pro-inflammatory mechanisms involving immune cell mobilization and local cytokine storms; nuclear factor-kappa B (NF-κB) and C-reactive protein are used as inflammatory biomarkers to assess the risk of ischemic damage [16].

Studies show that chronic hyperglycemia and dyslipidemia in diabetes increase the intravascular production of reactive oxygen species (ROS) by activating various ROS-producing enzymes. Xanthine oxidase (XO), enzymes of the mitochondrial respiratory chain, nicotinamide adenine dinucleotide phosphate (NADPH) oxidases, cyclooxygenase, and endothelial nitric oxide synthase (eNOS) have been reported to play a role in the pathogenesis of atherosclerosis in diabetes [19,20].

The ROS-sensitive factors, such as protein kinase C (PKC), a metabolic gene implicated in redox balance, and thioredoxin-interacting protein (TXNIP), also enhance the production of glyco/lipoxidation end-products, including AGEs and oxidized LDL, which result in increasing intravascular inflammation and leukocyte recruitment that further contributes to endothelial dysfunction [21,22,23].

Several clinical studies indicate that diabetic retinopathy is, in part, associated with the presence of atherosclerosis cardiovascular disease [24,25]. The combination of hyperglycemia, inflammation, and adiposity supports the pathogenetic links between diabetes, hypertension, and atherosclerotic cardiovascular disease; however, the underlying mechanism remains unclear.

### 2.2. Inflammation in Type 2 Diabetes

Inflammation is the innate immune system’s physiological reaction to maintain a constant internal setting while being exposed to continuously changing environmental pressure. Initial causes of those changes may originate from infectious, immunological, or chemical causes and/or metabolic dysfunction. The purpose of the inflammatory response is to reduce the causes of tissue injury, induce wound healing, and restore tissue homeostasis. The immune system orchestrates a cascade of inflammatory pathways and mechanistic effects to eradicate the causative agent [11].

Observational studies from decades ago revealed the first evidence for the association between inflammation and diabetes (e.g., administration of high doses of sodium salicylate led to decreased glycosuria in patients with suspected or definite diabetes) [26,27]. Later studies on the role of inflammation in diabetes revealed that this hypoglycemic action was related to the inhibition of serine kinase IkappaB kinase-beta (IKKbeta), which correlates with the post-receptor action of insulin [28].

In 1993, Hotamisiligil et al. determined the role of tumor necrosis factor-alpha (TNF-α) in insulin resistance and diabetes, demonstrating a correlation between inflammation and diabetes [29]. Elevated concentrations of markers and mediators of inflammation and acute-phase reactants, including C-reactive protein, interleukin (IL)-6, and plasminogen activator inhibitor-1, confirmed the epidemiologic associations between inflammation and obesity and type II diabetes [30,31,32,33,34]. Recent studies further evidenced the role of inflammation in diabetes initiation and progression [8,35].

Insulin resistance leads to both hyperglycemia and low-grade inflammation. Excess dysfunctional adipose tissue produces pro-inflammatory cytokines such as TNF-α and IL-6, while adiponectin release reduces known anti-inflammatory cytokines [36]. TNF-α and IL-6 are also related to elevated C-reactive protein levels [37]. These changes induce a state of systemic insulin resistance and low-grade inflammation [36]. In patients without diabetes, elevated C-reactive protein levels are related to the future development of insulin resistance and type 2 diabetes [38,39,40,41]. Obesity and metabolic syndrome have been reported as conditions in which subacute chronic inflammation is a common and potentially unifying mechanistic cause and is accompanied by activation of at least two major inflammatory pathways: stress-activated Jun N-terminal kinases and the transcription factor NF-κB [8,42,43,44,45,46,47]. Insulin resistance induced by hyperglycemia and obesity also converges with inflammation and immune modulation in the insulin receptor substrate 1 (IRS-1)/mitogen-activated protein kinase (MAPK)/Akt/phosphatidylinositol-3 kinase (PI3K) and NF-κB/IL-1β/TNF-α/interferon-γ (IFN-γ) pathways [18,43].

Regarding AGE accumulation in diabetes, we found that AGEs induce the activation of transcription factors NF-κB and specificity protein 1 (SP1) in retinal neurons [18]. This suggests that AGEs enhance RAGE gene expression in retinal neurons by activating NF-κB and SP1, leading to the production of pro-inflammatory cytokines and causing inflammation [18].

Insulin resistance is compensated by the hypersecretion of insulin in beta cells in the early stages of diabetes, and the clinical course of the disease occurs when pancreatic functional reserves eventually cannot cope with the required insulin secretion [48,49,50]. However, abnormal insulin sensitivity precedes the clinical diagnosis of diabetes by up to 15 years [51]; thus, the mechanisms that form the basis of insulin resistance should be investigated [52]. Moreover, studies have reported that other organs, such as the liver, neural system, skeletal muscle, and visual system, participate in metabolic homeostasis and the inflammatory state in type 2 diabetes, confirming systemic inflammation as a part of diabetes [53,54,55,56,57].

Changes in the liver and muscles have long been recognized as major contributors to systemic insulin resistance occurrence, as fat accumulation in the liver is a major factor in reduced hepatic insulin sensitivity, resulting in fasting hyperglycemia. Fat accumulation increases tissues’ insulin resistance and further pancreatic fat accumulation results in beta cell dysfunction [58,59]. Such conditions as metabolic syndrome, hypertension, and dyslipidemia, are associated with elevated concentrations of inflammatory biomarkers, which are predictive of the development of insulin resistance [58,59]. Visceral white adipose tissue (abdomen and upper body) appears to be the major source of inflammatory markers in type 2 diabetes (cytokines, TNF-alpha, IL-1, IL-6, IL-10, leptin, adiponectin, monocyte chemoattractant protein, angiotensinogen, resistin, chemokines, etc.) [60,61]. Later macrophages and immune cells (B cells and T cells) infiltrating adipose tissue trigger local and systemic chronic low-grade inflammation by producing chemokines and cytokines. This low-grade inflammation indicates a link between obesity, insulin resistance, and diabetes [62].

#### 2.2.1. Role of Inflammation in Diabetic Retinopathy

For decades, diabetic retinopathy was considered only a microvascular complication, but the retinal vessels are connected with and controlled by glia and neurons, which are found to be affected prior to vascular lesions [15,18,63,64]. Recently, in diabetic retinopathy studies has emerged a concept of the neurovascular unit, which includes the ganglion cells and the glial cells as neuronal parts and the endothelial cells and the pericytes as vascular components. The concept of the neurovascular unit reflects the interconnection of the blood flow regulation on the glial cells, pericytes, and neurons, as well as their reciprocal dependence on vascular support [65], confirming that neurodegeneration precedes microvascular abnormalities.

Recent studies have shown that the expression of cytokines (IL-1β, IL-6, IL-8, TNF-α, and monocyte chemotactic protein 1 (MCP-1)), chemokines (Cylco-oxygenase-2 (COX-2)), and other factors (vascular endothelial growth factor (VEGF), platelet-derived growth factor (PDGF), insulin-like growth factor (IGF-1), basic fibroblast growth factor (bFGF), and hepatocyte growth factor (HGF)) is increased in diabetes and that the levels of these molecules are particularly elevated in the ocular fluid of diabetic patients [66,67,68,69]. The accumulation of these cytokines is believed to lead to early neuronal cell death and play a crucial role in the angiogenesis, and increased expression of VEGF, contributing to the development of proliferative diabetic retinopathy [70]. 

It is known that Müller glial cells play a central role in retinal metabolism, and they are highly sensitive to metabolic changes associated with diabetes. Upregulation of glial fibrillary acidic protein (GFAP) by Müller glial cells is one of the early signs of retinal metabolic stress, which was shown in animal models and tissues from diabetic patients with no to mild nonproliferative diabetic retinopathy [70,71,72]. Consequently, the resident immune cells in the retina, called microglia, also become activated and start to produce pro-inflammatory mediators, exacerbating neuroglial and vascular dysfunction [73].

It has been proposed that microglial activation is the main mechanism by which neuroinflammation starts in response to that activation and production of inflammatory mediators [64].

The initial step for the inflammatory response in a diabetic retina may be due to metabolic change. Cell death was proposed as one of the causes. Retinal cell death in diabetic retinopathy happens via apoptosis resulting in mitochondrial damage. This is causing the upregulation of non-coding RNAs (mt-ncRNAs) [74]. Diabetic patients have increased circulating expressions of several LncRNAs, including Lnc*MALAT1* and Lnc*NEAT1* [75].

Factors that contribute directly or indirectly to increased inflammation are summarized in Figure 1.

#### 2.2.2. Inflammatory Pathways in the Neurovascular Cells of Diabetic Retinopathy

As mentioned above, glial cell (Müller cells, astrocyte, microglia) abnormalities are thought to be a trigger of orchestrated pathological changes in cells that consist of the neurovascular unit in diabetic retinopathy (Figure 1). Although most retinal cells are believed to die by apoptosis in early diabetic retinopathy, growing evidence has been indicating that various types of cell deaths are found in diabetic animal models, such as necroptosis, pyroptosis, or ferroptosis [76]. These regulated necrotic cell deaths are, in part, induced by inflammation and accelerate the inflammatory reaction via the distribution of damage-associated molecular patterns (DAMPs) in retinal tissues [76]. The cellular and molecular mechanisms underlying inflammatory reaction, including the distribution of DAMPs in diabetic retinopathy, are shown in Figure 2. The possible association between oxidative stress and neuroinflammation is displayed in the figure.

### 2.3. Nutritional Consequences of Hyperglycemia

Another important consequence of hyperglycemia, altered concentrations of trace elements, leads to changes in nutritional status. Minerals and trace elements are essential for many biochemical reactions, e.g., as stabilizing components of enzymes and proteins and cofactors for enzymes [77]. They bind to cell membrane receptors or change the shape of the receptor, thus regulating crucial biological processes [78]. These essential micronutrients have important physiological implications and are directly associated with diabetes mellitus [78,79].

Many studies have shown a relationship between altered metabolism of trace elements and the course of diabetes mellitus, and patients with diabetes mellitus were reported to have altered iron, copper, zinc, and magnesium metabolisms [77,78,79]. Some trace elements, such as chromium, magnesium, vanadium, zinc, manganese, molybdenum, and selenium, are reported to potentiate insulin action [80]. Proposed mechanisms of insulin action enhancement by trace elements include activating the insulin receptor site [81], serving as cofactors or components for enzyme systems involved in glucose metabolism [82], increasing insulin sensitivity, and acting as antioxidants to prevent tissue peroxidation [83].

Hyperglycemia alters the functions of vital trace elements such as zinc [84], which is required for insulin synthesis, storage, and the structural integrity of insulin. Insulin is secreted as zinc crystals [85]. Zinc has an important role in modulating the immune system, and its dysfunction in diabetes mellitus may be partly related to zinc status [85,86]. 

Magnesium plays an important role in the phosphorylation reactions of glucose and its metabolism. Magnesium deficiency has been implicated in insulin resistance, carbohydrate intolerance, dyslipidemia, and diabetes complications [87]. The association between diabetes mellitus and hypomagnesemia is compelling because of its wide-ranging impact on diabetic control.

Trace element deficiencies are associated, either directly or indirectly, with oxidative stress, eventually leading to insulin resistance or diabetes, and appear to be additional risk factors for diabetes development and progression [88]. These deficiencies also contribute to the pathogenesis of diabetes mellitus and its complications.

## 3. Future Perspectives for Treatment of Diabetes and Diabetic Retinopathy

Various modifiable (e.g., environmental, nutrition, and lifestyle) and non-modifiable (genetic) risk factors are believed to influence the onset of diabetes, its development, and its course. The continuous presence or coexistence of these risk factors appears to trigger underlying molecular and cellular pathways that can lead to loss of tissue homeostasis and tissue dysfunction. This condition may develop ahead of cellular changes that cause tissue disorders if not counterbalanced by the immune system and preventive measures such as a healthy diet and lifestyle. In these cases, the disease finally manifests clinically, and medical intervention may be required.

Researchers must clarify the molecular and cellular pathways along with the underlying multifactorial causes of diabetes to establish effective measures with fewer adverse effects, which is difficult and demanding. Disruption of the homeostatic balance via interconnected cascade reactions and a complex network of intertwined molecular processes overlapping redox product clearance, stress and damage response, glycemic stress, and inflammation demonstrates the lack of holistic approaches in the currently available therapeutic strategies. These approaches include agent-one enzyme-based molecular regulators and treatments compared with a holistic approach to treating diabetes as a multifactorial disorder. Such novel approaches as complex measures of anti-inflammatory diets [89,90], modulation of the microbiome [91], the establishment of anti-inflammatory therapies, immune modulation [92], peptides [93], and mRNA therapies [94] are gaining interest in new studies.

### 3.1. Therapeutic Options of Anti-Inflammation for Diabetic Retinopathy

Recent studies indicate that several anti-inflammatory therapies ameliorate the pathogenesis of diabetic retinopathy in diabetic models in vivo and in vitro, including IMD-0354, a specific NF-κB blocker [95], GSK-872, specific necroptosis inhibitor [96], hsa_circ_0000047 which targets miR-6720-5p/CYB5R2 axis [97], topical administrated NADPH oxidases 4 inhibitor, GLX7013114 [98], a non-steroid mineralocorticoid receptor, finerenone [99], and miRNA-124 [100]. Although there are a variety of diabetic models, including rodents and Drosophila, which are available worldwide, no ideal models of diabetic retinopathy have been established yet [101,102]. Thus, researchers must carefully select animal models of diabetic retinopathy fitted with their own purpose. Olivares et al. summarizes diabetic animal models in their review. Researchers can refer to the review before setting up the experiments to determine whether researchers focus on the early or the late stage and neuronal abnormalities or vascular abnormalities [101].

Minocycline, one of the tetracycline antibiotics, has been reported to reduce the inflammatory cytokines, including IL-1β and to protect from retinal neuronal cell death by reducing caspase-3 activation via inhibiting microglia activation in diabetic retinopathy [103,104]. However, the effects of minocycline cannot be beyond the currently available diabetic therapies such as insulin or pioglitazone [105], and thus, minocycline may be limited for clinical use for the treatment of diabetic retinopathy. Although several candidates of anti-inflammatory therapies for diabetic retinopathy are in basic science, there are few clinical studies for determining the effects of anti-inflammatory therapies for preventing the progression of diabetic retinopathy. Thus, further clinical trials for evaluating the efficacies of anti-inflammatory therapies in patients with diabetic retinopathy should be performed in the future.

### 3.2. Anti-Inflammatory Therapies for Diabetic Macular Edema

Although preventing the progression of diabetic retinopathy and anti-inflammatory therapies seem not to be beyond the current available diabetic therapies, diabetic macular edema is a representative and an ideal target of anti-inflammatory therapies because inflammation is critically associated with the pathogenesis of diabetic macular edema. Although it is a personal opinion, the possible trigger of the inflammatory reaction of the development of diabetic macular edema is glycation and accumulation of AGEs. Glycated lipids or proteins and accumulated AGEs in the macula can be recognized as foreign matters by immunocompetent cells, and the inflammatory reaction is induced, which results in increased vascular permeability. A recent meta-analysis indicates that 4% of diabetic patients have sight-threatening clinically significant macular edema [106]. A previous randomized, sham-controlled clinical study indicates that dexamethasone intravitreal implant significantly improved best-corrected visual acuities in eyes with diabetic macular edema for three years [107]. Koc et al. compared the efficacy of monthly anti-VEGF versus dexamethasone implants after three doses of anti-VEGF therapy [108]. There is no significant difference in the efficacy at six months between monthly anti-VEGF injection and switching to a dexamethasone implant after three doses of anti-VEGF for patients with diabetic macular edema [108]. They recommend dexamethasone implants compared to anti-VEGF because of fewer injections and lower cost of dexamethasone implants than monthly anti-VEGF injections [108]. In Japan, sustained-release steroid agents such as dexamethasone implants have not been approved yet. Therefore, on behalf of sustained-release steroid agents, we usually select sub-Tenon triamcinolone acetonide injection or intravitreal triamcinolone acetonide injection for the treatment of diabetic macular edema. Our previous study indicates that switching from anti-VEGF to triamcinolone acetonide injections showed significant therapeutic effects in eyes with refractory diabetic macular edema treated with anti-VEGF injection [109]. Anti-VEGF can only reduce the level of VEGF in the eye, but other cytokines such as IL-6, IL-8, IL-10, or MCP-1 cannot be reduced by anti-VEGF injection. However, steroid therapies can reduce other cytokines described above, and in some situations, steroid therapies can be beyond anti-VEGF injection [110]. For example, Kandasamy et al. performed a prospective, randomized clinical trial to compare the effects of intravitreous bevacizumab and triamcinolone when administrated at the time of cataract surgery accompanied by diabetic macular edema [111]. The results of the clinical trial indicated that when administrated at the time of cataract surgery accompanied by diabetic macular edema, only the triamcinolone acetonide injection group resulted in a sustained reduction in the central macular thickness and the intravitreous bevacizumab injection group could not reduce the central macular thickness during the six-months follow-up period [111]. Although it is heterogenous, a recent meta-analysis to compare outcomes of cataract surgery combined with either anti-VEFG injection or dexamethasone implant in patients with diabetic macular edema suggests that the group of cataract surgery combined with dexamethasone implant seems to provide better outcomes compared with the group of anti-VEGF injection [112]. Taken together, anti-inflammatory therapies are effective for the treatment of diabetic macular edema, and thus, inflammation is, in part, associated with the pathogenesis of diabetic macular edema. Persistent diabetic macular edema may cause permanent vision decrease because of photoreceptor cell death. Thus, anti-inflammatory therapies for diabetic macular edema may be one of the neuroprotective therapies in a broad sense.

### 3.3. Anti-Hyperlipidemic Therapies for Diabetic Retinopathy

There are several common signaling pathways utilized by both dyslipidemia and diabetes, including AMP-activating protein kinase, peroxisome proliferator-activated receptor (PPAR) protein family signaling, and inflammatory markers [113]. Recent diabetic animal models studies pointed to peroxisome proliferator-activated receptor alpha (PPARα) activation in cellular metabolism and inflammation by administration of oral fenofibrate (pemafibrate), which was proposed as a promising target for diabetic retinopathy [114]. Fenofibrate, by upregulating the production of molecules for fatty acid transport and β-oxidation through the activation of PPARα, is able to reduce free fatty acids levels [114,115]. Additionally, fenofibrate has the potential to induce an increase in the synthesis of high-density lipoprotein cholesterol and apolipoproteins. As in the therapy for diabetic retinopathy, administration of fenofibric acid (a metabolite of fenofibrate) reduced ganglion cell death and preserved amplitudes in oscillatory potentials and implicit time of b-wave in diabetic db/db mice [116]. There are two randomized, placebo-controlled, multicentral clinical trials demonstrating the therapeutic effect of fenofibrate on the progression of diabetic retinopathy [117,118]. Both clinical trials indicate that fenofibrate significantly reduced the progression of diabetic retinopathy (*n* = 9795 and 10,251, respectively) [117,118]. 

Pemafibrate is a new selective PPARα modulator, suggested by Kowa Company, Ltd. as a more effective alternative to fenofibrate, demonstrated a significant reduction in retinal neovascularization in a murine model of oxygen-induced retinopathy [119]. In experiments, pemafibrate therapy shows hope as a potential therapy for diabetes and diabetic retinopathy. Further clinical trials should be performed to establish the early treatment for diabetic retinopathy in clinical practice.

### 3.4. Pyroptosis; A New Therapeutic Target for the Treatment of Diabetic Retinopathy

Pyroptosis is an inflammatory-related programmed cell death and is involved in the neurovascular impairment of diabetic retinopathy [120]. Briefly, after binding the first signals, such as pathogens or cytokines, with the Toll-like receptor, NF-κB signaling is activated, which results in upregulating inflammatory genes, including pro-IL-1β, pro-IL-18, and NLRP3 (Figure 2). The second signals, such as NLRP3 agonists, induce ROS production as a result of mitochondrial damage. NLRP3 agonists also increase potassium efflux via the K channel linked with the P_2_X_2_ receptor, which leads to activated NLRP3 inflammasome followed by caspase-1 activation (Figure 2). Finally, excessive potassium efflux and caspase-1 activation lead to pyroptosis by cleavage of gasdermin D [110,120]. A recent study indicates that cell pyroptosis of diabetic retinopathy was found in specifically retinal ganglion cells and a flavonoid extracted from the traditional Chinese medicine, scutellarin protected retinal ganglion cell pyroptosis in diabetic retinopathy in vivo via inhibition of caspase-1, gasdermin D, NLRP3, IL-1β and IL-18 [121]. Ginsenosides are steroid-like saponins and are the main active components of ginseng, a well-known traditional Chinese medicine. Growing evidence shows that ginsenoside has significant effects in ameliorating the pathological changes of diabetes and its complications via inhibition of the pyroptosis-associated inflammasome pathway [122]. Li et al. indicate that MiR-200c-3p regulated high glucose-induced pyroptosis of human retinal microvascular endothelial cells by targeting SLC03A7 [123]. Gu et al. suggest that microRNA-192 represses retinal pigment epithelium cell pyroptosis induced by high glucose via regulation of the FTO-α-keratoglutarate-dependent dioxygenase/NLRP3 signaling pathway [124]. Ma et al. demonstrate that the knockdown of transient receptor potential channel 6 reduced inflammation and cell pyroptosis in high glucose-induced retinal Müller cells, and transient potential channel 6 may be a potential target for the treatment of diabetic retinopathy [125]. Taken together, anti-pyroptosis therapies may become new therapeutic options for diabetes and diabetic retinopathy in the future.

Anti-inflammatory therapies for preventing the progress of diabetic retinopathy have not been established yet, but in some pathological conditions, such as diabetic macular edema or after surgical management, anti-inflammatory therapies may be superior to other therapies for patients with diabetic retinopathy. However, the pathological mechanisms involved in the inflammation in the neurovascular cells of diabetic retinopathy are not elucidated yet. Further basic and clinical studies are heartily welcome to establish novel anti-inflammatory therapies for diabetic retinopathy.

## 4. Conclusions

The inflammation-nutrition interaction has been described as a “double-edged sword”, where metabolic stress induces inflammation, and inflammation disrupts metabolic homeostasis in different tissues [126]. This interaction is as complex as the underlying metabolic system itself, hindering effective therapeutic strategies as a multi-pathway approach to managing cardiovascular disease, hyperglycemia/diabetes mellitus, obesity, and associated disorders. Glycation, inflammation, and catabolism are the main pathological processes that significantly affect our health. They are closely interconnected and may trigger many undesirable reactions leading to a variety of diseases, from diabetes to oncology.

The potential role of primary physicians and endocrinologists in the screening and management of diabetic retinopathy, particularly in cases where patients may not have access to specialized ophthalmology care, may not be only the blood sugar control but also the control of hyperlipidemia and blood pressure, and the monitor of atherosclerosis cardiovascular disease. Dietary controls, i.e., recommendation of low-AGE foods and anti-oxidative nutrients, may be one of the options for the primary management of diabetic patients [63].

Although new strategies such as immunomodulation, peptide therapeutics, and antisense miRNAs are being introduced, more detailed studies and revisions in the understanding of the pathophysiology of age-related diseases and diabetic retinopathy are still required for effective prevention, management, and development of novel interventions.

Personalized approaches might guide a new generation of clinical studies to provide more effective solutions for managing diabetes mellitus and diabetic retinopathy.

## Figures and Tables

**Figure 1 ijms-24-09166-f001:**
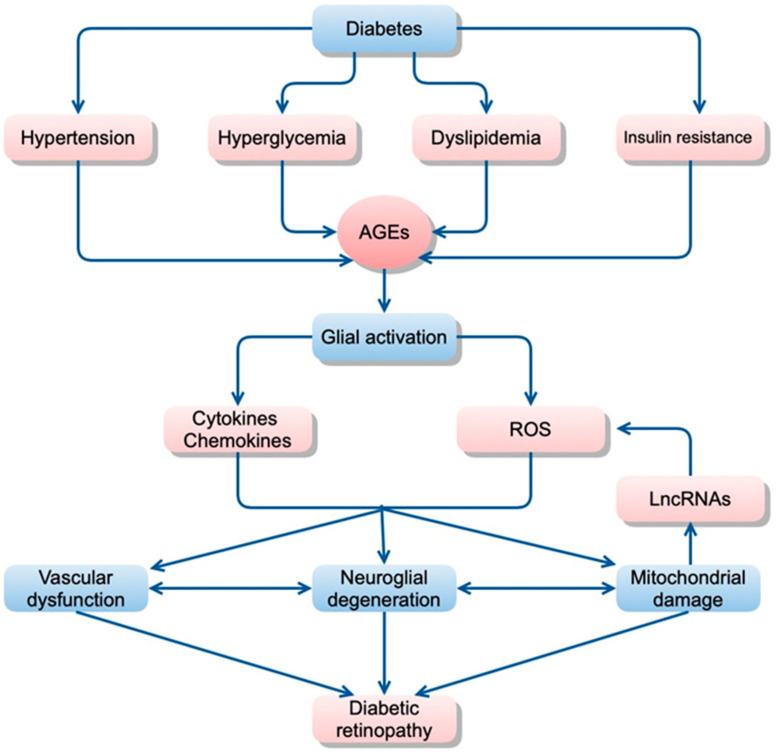
Hypothetical scheme of the pathogenesis of diabetic retinopathy. Metabolic changes, including hyperglycemia, dyslipidemia, and insulin resistance, resulting in glial dysfunction, which induces inflammation, aberrant signaling of trophic factors, and metabolic dysregulation, all leading to neuronal apoptosis. AGE accumulation may be a critical trigger for inducing inflammation in diabetic retinas probably because accumulated AGEs may be recognized as foreign materials by immunocompetent cells, which results in inducing inflammatory reactions in the retina. (AGE, advanced glycation end-products; ROS, reactive oxygen species; LncRNAs, long non-coding RNAs).

**Figure 2 ijms-24-09166-f002:**
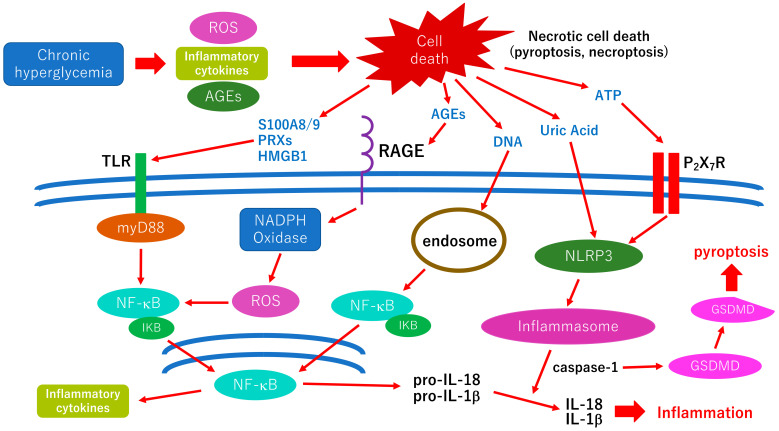
Hypothetic scheme of the cellular and molecular mechanisms underlying inflammatory reaction in diabetic retinopathy. Retinal cell death in diabetic retinopathy is thought to be induced by several diabetic stress, including oxidative stress via the production of ROS, inflammatory cytokines, and AGE accumulation. Retinal cells in diabetic retinopathy are believed to die by apoptosis or regulatory necrotic cell death, such as pyroptosis, necroptosis, or ferroptosis. Necrotic cell death distributes varieties of DAMPs (ATP, uric acid, DNA, AGEs, S100A8/9, peroxiredoxins (PRXs), high mobility group box-1 (HMGB1), etc.) in the retinal tissue. These DAMPs are recognized by each pattern recognition receptor and activate each intracellular pathway, which results in inducing further inflammatory reactions. In the AGE-RAGE axis, NADPH oxidase is activated, followed by increasing the production of ROS. The increase of intracellular ROS activates the NF-κB pathway, which results in releasing inflammatory cytokines. Vascular cell death should be involved in vascular permeability, and the development of diabetic macular edema, and neuronal cell death should be associated with irreversible vision decrease in diabetic retinopathy. Thus, anti-inflammatory therapies may be effective in preventing both vascular and neuronal abnormalities in diabetic retinopathy. ROS: reactive oxygen species; AGEs: advanced glycation end-products; TLR: Toll-like receptor; RAGE: receptor for AGE; P_2_X_7_R: P_2_X_7_ receptor; Myd88: myeloid differentiation primary response gene 88; NADPH: nicotinamide adenine dinucleotide phosphate; NLRP3: Nod-like receptor family pyrin domain containing 3; IκB: inhibitor of κB; GSDMD: Gasdermin D.

## Data Availability

Not applicable.

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
