# Peer review of "Diabetic Neuropathy of the Retina and Inflammation: Perspectives"

_ijms, 2023, doi:10.3390/ijms24119166_

Round 1

Reviewer 1 Report

The paper " Diabetic Neuropathy of the Retina and Inflammation: Perspectives" provides a comprehensive review of the current understanding of the role of inflammation in the development and progression of diabetic retinopathy. The paper is well-structured, and the arguments are clearly presented, with appropriate use of references to support the authors' claims. However, there are a few areas that could be improved or clarified:

·         The paper mainly focuses on the role of inflammation in the development of diabetic retinopathy. However, it would be beneficial to include a section discussing the potential therapeutic strategies targeting inflammation in the treatment of diabetic retinopathy.

·         While the authors discuss the role of inflammation and neurovascular dysfunction in the pathogenesis of diabetic retinopathy, they do not provide much detail on the specific inflammatory pathways or neurovascular mechanisms involved. Providing more information on the specific molecular and cellular pathways underlying these processes would be helpful.

·         Please add several potential therapeutic interventions for diabetic retinopathy, including neuroprotective agents and anti-inflammatory drugs. However, it is not clear how advanced these interventions are in terms of development and clinical testing. It would be helpful to provide more information on the current state of research and clinical testing for these interventions.

·         The authors should provide a detailed description of the cellular and molecular mechanisms underlying inflammation in diabetic retinopathy. It would be helpful to provide more information on the clinical implications of these mechanisms. For example, how do these mechanisms relate to the clinical manifestations of diabetic retinopathy, such as retinal neovascularization or macular edema?

·       It would be beneficial to discuss the potential role of primary care physicians and endocrinologists in the screening and management of diabetic retinopathy, particularly in cases where patients may not have access to specialized ophthalmology care.

·        The prevalence of diabetic retinopathy is increasing worldwide.  Provide detail on the specific factors contributing to this trend. It would be helpful to provide more information on the underlying demographic, environmental, and lifestyle factors that may be driving the increasing prevalence of diabetic retinopathy.

·         Discuss the potential mechanisms underlying neuroinflammation and neurodegeneration in diabetic retinopathy.  Potential role of other factors, such as oxidative stress, in the pathogenesis of the disease. It would be beneficial to discuss the possible interplay between these different mechanisms and how they may contribute to disease progression.

·         Overall, the paper provides valuable insights into the role of inflammation in diabetic retinopathy and is well-structured and clearly presented. However, there are a few areas where additional clarification or improvement would be beneficial. 

Regarding the quality of the English language, the paper is well-written and free of major grammatical errors or spelling mistakes. However, there are a few areas where the language could be improved to enhance clarity and readability. For example, there are a few instances where sentences are overly long and complex, which can make it difficult for readers to follow the argument. Additionally, there are a few instances where word choice could be improved to enhance precision and avoid ambiguity. Overall, the language is of a high standard but could benefit from some minor revisions.

Author Response

Editorial Office

To the Editor

We submit our revised manuscript titled,

Diabetic neuropathy of the retina and inflammation: perspectives

to be considered for publication in International Journal of Molecular Sciences. Our responses to the reviewers’ comments are presented below.  We would appreciate your kind review again.

Sincerely,

Toshiyuki Oshitari, MD, Ph.D.

Department of Ophthalmology and Visual Science

Chiba University Graduate School of Medicine

Inohana 1-8-1, Chuo-ku, Chiba 260-8670, Chiba, Japan.

TEL:81-43-226-2124

FAX:81-43-224-4162

e-mail:[email protected]

Reviewer 1

  1. It would be beneficial to include a section discussing the potential…

Answer. We have included the section “Therapeutic options of anti-inflammation for diabetic retinopathy” in the discussion.

  1. They do not provide much detail on the specific inflammatory pathways…

Answer. We have added a new figure (Figure 2) to display specific inflammatory pathways.

  1. Please add several potential therapeutic interventions for diabetic retinopathy…

Answer. We have added potential therapeutic interventions in the treatment sections.

  1. The authors should provide a detailed description of the cellular and the molecular mechanisms…

Answer. We have added a new figure (Figure 2) for providing a detailed description of the cellular and the molecular mechanisms underlying inflammation in diabetic retinopathy.

  1. It would be beneficial to discuss the potential role of primary physicians…

Answer. We have added the potential role of primary physicians and endocrinologists in the screening and management of diabetic retinopathy.

  1. The prevalence of diabetic retinopathy is increasing…

Answer. We have added the estimated prevalence of diabetes and diabetic retinopathy in the introduction.

  1. Discuss the potential mechanisms underlying neuroinflammation…

Answer. We have added a new figure (Figure 2) and discuss the potential mechanisms underlying neuroinflammation and neurodegeneration in diabetic retinopathy including oxidative stress.

Reviewer 2.

  1. Lines 196-198…

Answer. We have added the reference.

  1. All non-standard abbriviation….

Answer. We have checked the abbreviation in the whole manuscript again.

Reviewer 3.

  1. The author added more recent publications on the...

Answer. We have added new publications including all references you suggested.

  1. I would add information on pathways…

Answer. We have added a new figure (Figure 2) to display the cellular and the molecular pathways underlying inflammation in diabetic retinopathy.

Reviewer 2 Report

The manuscript titled, "Diabetic Neuropathy of the Retina and Inflammation: Perspectives" by Bikbova et al., is a valuable contribution to the role of inflammation to the development of diabetic complications and deserves publication after a few minor revisions as follows:

1) lines 196-198: Is a reference missing? Please review this sentence and the sentence that follows (line 199-202).

2) All non-standard abbreviations/acronyms should be written out in full when first used and followed by the abbreviated form in parentheses.

Author Response

(The authors gave the same response as above.)

Reviewer 3 Report

I believe it will be more beneficial if the author added more recent publications on the neural vascular unit damage during DR. Also, I would add information on pathways that are specifically activated in DR which induce neuronal damage and does not directly impact vascular health in the early phase of DR manifestation.

It would be interesting to go a little into clinical research and animal research on DR too. I would refer the authors to the below publications

  • PMID: 33828528; PMID: 28836097; PMID: 33582248 etc. 

  I think since the author brings up cardiovascular link to DR it would be nice to describe some of the research found on this topic. PMID: 33902673; PMID: 36407461.

It would be nice to include some currently ongoing clinical therapies for neuronal aspects of DR.

The English usage is fine in this manuscript

Author Response

(The authors gave the same response as above.)
